# Right Anterior versus Right Transaxillary Access for Minimally Invasive Aortic Valve Replacement: A Propensity Matched Competitive Analysis

**DOI:** 10.3390/jcm13040985

**Published:** 2024-02-08

**Authors:** Ali Taghizadeh-Waghefi, Sebastian Arzt, Lisa Wenzel, Asen Petrov, Manuel Wilbring, Klaus Matschke, Utz Kappert, Konstantin Alexiou

**Affiliations:** 1Medical Faculty “Carl Gustav Carus”, TU Dresden, 01307 Dresden, Germany; sebastian.arzt@herzzentrum-dresden.com (S.A.); asen.petrov@herzzentrum-dresden.com (A.P.); manuel.wilbring@herzzentrum-dresden.com (M.W.); klaus.matschke@herzzentrum-dresden.com (K.M.); utz.kappert@herzzentrum-dresden.com (U.K.); konstantin.alexiou@herzzentrum-dresden.com (K.A.); 2Center for Minimally Invasive Cardiac Surgery, University Heart Center Dresden, 01307 Dresden, Germany

**Keywords:** aortic valve replacement, minimally invasive surgery, transaxillary, MICLATS

## Abstract

(1) Background: Right anterior thoracotomy (RAT-AVR) has been the sole established sternum-sparing technique for minimally invasive aortic valve replacement (MICS-AVR) thus far. Nevertheless, transaxillary access, known as Minimally Invasive Cardiac LATeral Surgery (MICLATS-AVR), represents the latest and innovative advancement in sternum-sparing MICS-AVR access routes. In this study, procedural and clinical outcomes of a substantial transaxillary MICS-AVR cohort are compared to those of a RAT-AVR control group; (2) Patients and Methods: This retrospective study included 918 consecutive patients who underwent MICS-AVR at our facility between 2014 and 2022. This cohort was divided into two surgical access-related groups: RAT-AVR (*n* = 492) and MICLATS-AVR (*n* = 426). Procedural data, operative morbidity, and mortality were compared between groups. Further analysis was performed using propensity score matching; (3) Results: After matching, 359 pairs of patients were included and analyzed. There were no notable differences observed between the two groups regarding major adverse cardio-cerebral events. Despite longer cardiopulmonary bypass time in the MICLATS-AVR group (63.1 ± 20.4 min vs. 66.4 ± 18.2 min; *p* ≤ 0.001) the skin-to-skin time (129.4 ± 35.9 min. vs. 126.5 ± 29.8 min.; *p* = 0.790) and the aortic cross-clamp time was comparable between both groups (41.9 ± 13.3 min. vs. 43.5 ± 14.4 min.; *p* = 0.182). The overall hospital stay was significantly shorter in the MICLATS-AVR cohort (9.7 ± 5.2 days vs. 9.2 ± 4.5 days; *p* = 0.01). Both groups were comparable in terms of postoperative morbidities. However, significantly lower rates of postoperative impaired wound healing were noted in the MICLATS-AVR group (11.7% vs. 3.9%, *p* < 0.001); (4) Conclusions: In comparing MICLATS-AVR and RAT-AVR, our study found MICLATS-AVR to be at least as safe and time-efficient as RAT-AVR, with no significant differences in MACCE. MICLATS-AVR showed a shorter hospital stay and lower postoperative wound issues, indicating its feasibility and safety as an alternative. Notably, MICLATS-AVR is sternum- and bone-sparing, preserving the right mammary artery, and facilitates combined procedures like multiple valve surgeries.

## 1. Introduction

Surgical aortic valve replacement (SAVR) has been considered the gold standard for the treatment of severe aortic valve stenosis for many years and has traditionally been performed via median sternotomy. However, in recent years, there has been increasing interest in less invasive approaches to SAVR, known as minimally invasive surgical aortic valve replacement (MICS-AVR). This growing interest was driven, in part, by the fact that the development of interventional trans-aortic catheter-assisted valve implantation (TAVI) added another less invasive procedure to the therapy strategies of aortic valve stenosis. Across all surgical risk categories, TAVI has demonstrated non-inferior mortality outcomes compared to SAVR in the short to medium term [1,2,3,4,5]. The picture that emerges in current clinical practice is that the TAVI procedure and MICS-AVR techniques have opposite goals. While the TAVI procedure is expanding its clinical applicability from high- to low-risk patients, the MICS-AVR claims to extend its indication from low-risk to higher-risk patients. However, the MICS-AVR technique cannot be considered as a universally standardized surgical procedure. Rather, it constitutes an inclusive and overarching surgical concept encompassing a varied and diverse array of surgical approach methods and cannulation strategies.

The first step towards MICS-AVR was taken by Rao and Kumar in 1993, introducing a right anterolateral thoracotomy approach [6]. In 1996, Cosgrove and Sabik reported a conceptually pioneering approach to MICS-AVR in a small case series that used femoral cannulation to achieve greater surgical comfort and wider access to the surgical field despite reduced thoracic incision [7]. In an ongoing effort to develop access techniques that avoid full median sternotomy, partial sternotomy has emerged as the most used surgical approach for MICS-AVR. Thus, the proportion of partial sternotomies among isolated surgical aortic valve procedures is reported to currently be 39.7% in Germany [8]. To address this problem, alternative sternum-sparing intercostal access techniques have been proposed, allowing a SAVR procedure with direct vision via right anterolateral thoracotomy (RAT-AVR). Prior to recent developments, this access method stood as a unique and exclusive approach in its category. A more recent and innovative technique is the “single incision-direct vision” right lateral transaxillary access, known as Minimally Invasive Cardiac LATeral Surgery (MICLATS), which involves an incision of 4 to 5 cm in the right anterior axillary line [9]. This approach has shown promising initial outcomes compared to full median sternotomy [10]. However, the safety and efficacy of this approach require further investigation. In this study, we aim to compare the safety and efficacy of the two sternum-sparing methods of right lateral transaxillary access (MICLATS-AVR) with the widely used RAT for SAVR (RAT-AVR) in a large cohort of patients.

## 2. Materials and Methods

### 2.1. Inclusion and Exclusion Criteria

The primary objective of this study was to investigate adult patients who underwent isolated aortic valve replacement, comparing those who received MICLATS-AVR in the treatment group to those who underwent RAT-AVR in the control group. Patients who met any of the exclusion criteria, including those who underwent other combined procedures, had active or recent endocarditis or had previously undergone redo surgeries, were excluded from the study.

### 2.2. Study Design and Ethical Statement

The present single-center study constitutes a retrospective cohort analysis, evaluating consecutive patients who underwent MICS-AVR through right lateral transaxillary access. To ensure comparability, a 1:1 propensity score-matching technique was employed. The control group was obtained from a retrospectively analyzed cohort of consecutive patients who underwent MICS-AVR via right anterolateral thoracotomy. The inclusion and exclusion criteria were consistent for both groups of patients. The data were extracted retrospectively from the hospital’s electronic health records. The primary endpoints were defined as major adverse cardio-cerebral events (MACCE), including perioperative myocardial infarction, perioperative ischemic stroke, and 30-day mortality. Besides the main postoperative results related to significant adverse cardio-cerebral events, secondary postoperative outcomes involved a thorough evaluation of additional postoperative morbidities and complications. The research protocol was subject to review and approval by the local Ethics Board.

### 2.3. Patient Population

Over the period of 2014 to 2022, a total of 2171 patients underwent elective isolated primary surgical aortic valve replacement at our medical facility for degenerative aortic valve disease. However, 1253 patients who received full or upper partial sternotomy were excluded from this retrospective analysis, leaving a total of 918 patients who underwent isolated sternum-sparing MICS-AVR to be included in the study. The cohort of patients who underwent isolated sternum-sparing MICS-AVR was divided into two groups, RAT-AVR and MICLATS-AVR, based on the surgical approach used. The selection process of patients is visually depicted in a flowchart format in Figure 1.

### 2.4. Involved Surgeons

All MICS-AVR procedures were primarily conducted by four surgeons, each possessing expertise in the field of adult cardiac surgery, with all-over procedural volumes exceeding >500, >2500, >3000, and >6500 cases involving extracorporeal circulation (ECC), respectively.

Additionally, a structured modular training approach was implemented for residents undergoing systematic step-by-step instruction in specific components of the MICS-AVR procedure. This encompasses training on femoral cannulation techniques, principles of thoracotomy, aspects of thoracotomy closure, and exposure of the aortic valve through various aortotomy techniques adapted to the specific type of valve prosthesis to be implanted. Thus, in line with the institutionalization and conceptualization of the MICS-AVR techniques, residents are systematically assigned specific procedural components in a modular manner. This modular concept is designed to afford residents practical experience, enabling them to progressively undertake the operation themselves, initially under the supervision of a consultant and eventually autonomously.

### 2.5. Sternum-Sparing MICS-AVR Access Routes

The choice of minimally invasive surgical access route was based on the surgeon’s preference and the patient’s anatomical features assessed via electrocardiography-gated computed tomography angiography of the thorax, abdomen, and pelvis. All patients underwent general anesthesia. Intraoperative transesophageal echocardiography (TEE) was performed as a standard imaging protocol, irrespective of the surgical method employed. As part of the standard anaesthetic protocol for all sternum-sparing MICS-AVR, the airway was intubated with a double-lumen ventilation tube, allowing single-lung ventilation. A temporary transvenous pacing wire was inserted through a percutaneous introducer sheath for all patients undergoing MICS-AVR. Despite variations in the surgical entry site, a conventional surgical cut-down was utilized for the exposure of femoral vessels being cannulated under TEE guidance to establish ECC. Additionally, common to all MICS-AVR was the placement of the antegrade cardioplegia cannula in the ascending aorta and the implementation of a left ventricular vent line via the right superior pulmonary vein.

To employ the RAT-AVR, the patient is placed in a supine position. A 5 cm incision is made along the second intercostal space, followed by dissection of the pectoralis and intercostal muscles as well as the right mammary artery. Using an oscillating saw, the third rib is meticulously detached from the sternum in a wedge shape.

The MICLATS-AVR access is performed as previously described in earlier publications. In short, for the MICLATS-AVR, the patient is placed in the supine position with the right side of the chest slightly elevated by two pillows and the right arm raised and attached to an armrest of the operating table, which is called the javelin thrower’s position (Figure 2). A 5 cm incision is made along the right anterior axillary line, and subsequent dissection of the serratus anterior and intercostal muscle is carried out to access the third or fourth intercostal space [9,11].

### 2.6. Prosthesis Choice

The choice of prosthesis typically relied on the preferences of the operating surgeon, particularly in the context of MICS and in patients undergoing MICS-AVR. In these cases, detailed anatomical data for procedural planning were obtained through a high-resolution, full cardiac cycle computerized tomography scan using the TAVI-CT protocol. This preoperative information included measurements such as the distances from the annular plane to the chest wall, aortic annulus size, and the anticipated size of the implanted valve. This valuable data played a crucial role in the surgeon’s decision-making process. Due to the increasing evidence supporting the durability of the rapid deployment valves used, their utilization was unrestricted [12,13,14].

### 2.7. Statistical Analysis

The data for continuous variables were assessed for normal distribution using the Kolmogorov–Smirnov test with Lilliefors significance correction at a type I error rate of 10%. If the data were normally distributed, further analysis was conducted to evaluate variance heteroscedasticity using the Levene test with a type I error rate of 5%. Subsequently, if both normality and variance homogeneity were established, an independent two-sample *t*-test was applied to compare subgroups. If the data showed normality but lacked variance homogeneity, Welch’s *t*-test was utilized. For variables with non-normal distribution and those measured on ordinal scales, the Mann–Whitney U test was employed. Fisher’s exact test was used to compare dichotomous variables, while the chi-square test (exact or with Monte Carlo simulation) was utilized for the other categorical variables. 

To compare RAT-AVR versus MICLATS-AVR, subsets with reduced bias were obtained through propensity score matching (matching variables: EuroSCORE II, Society of Thoracic Surgeons Predicted Risk of Mortality [STS-PROM], age, sex, body mass index [BMI], preoperative left ventricular ejection fraction [LVEF], estimated creatinine clearance according to the Cockcroft–Gault equation [CRCL], preoperative New York Heart Association [NYHA] classification, diabetes mellitus, pulmonary arterial hypertension, coronary artery disease, peripheral occlusive arterial disease, and chronic obstructive pulmonary disease [COPD]). To achieve optimal pairings, a maximum permissible disparity of 0.08 was established between two given patients. As the type I error was not corrected for multiple testing, any inferences drawn from the results of inferential statistics serve only a descriptive purpose. Thus, the usage of the term “significant” in the study’s findings indicates solely a local p-value below 0.05 and does not imply any error probability less than 5%. The statistical analyses were conducted using the open-source R statistical software (version 4.1.2).

## 3. Results

### 3.1. Patient Baseline Characteristics

Between January 2014 and February 2022, a total of 918 patients underwent sternum-sparing MICS-AVR. These patients were categorized into two groups based on the surgical access routes for MICS-AVR: 492 underwent RAT-AVR, and 426 underwent MICLATS-AVR. No significant differences in baseline demographic characteristics, such as sex and age, were observed between the two study groups. However, the MICLATS-AVR group exhibited significantly higher body weight and, consequently, a higher body mass index. Additionally, the calculated creatinine clearance was notably higher in the MICLATS-AVR group. Notably, a significantly higher proportion of patients in the MICLATS-AVR group fell into NYHA class III and IV. In all other aspects, the additional baseline characteristics of the two groups were comparable. Furthermore, there were no significant differences between the groups concerning the EuroSCORE II risk stratification score. Nevertheless, the STS-PROM score indicated a slightly, yet statistically significantly, higher estimated 30-day mortality for the RAT-AVR group.

We matched 359 patients from each group within the entire cohort using propensity scores derived from the mentioned variables, thereby ensuring that both groups exhibited similar baseline characteristics. Following the propensity score matching, we observed a marked alignment in baseline characteristics between the two groups. The baseline characteristics of the pre-matched and propensity-matched groups are shown in Table 1.

### 3.2. Unadjusted Outcomes

#### 3.2.1. Procedural and Intraoperative Data

The results for this section are summarized in Table 2. Significantly more rapid deployment valves were implanted in the MICLATS-AVR group than in the RAT-AVR group (81.9% vs. 53.0%; *p* ≤ 0.001; Figure 3). Mechanical substitutes were infrequently implanted in both groups. The size of the implanted valve prosthesis was slightly larger in the MICLATS-AVR group (24.2 ± 2.1) compared to the RAT-AVR group (24.0 ± 1.9), and this difference reached statistical significance (*p* = 0.044). While there were no differences in skin-to-skin time between the two groups, it is worth noting that the MICLATS-AVR group had significantly longer cardiopulmonary bypass time (63.2 ± 24.5 vs. 66.4 ± 18.5; *p* ≤ 0.001) and aortic cross-clamp time (41. 9 ± 14.2 vs. 43.5 ± 14.4; *p* = 0.044) compared to the RAT-AVR group (Figure 4). 

#### 3.2.2. Postoperative Outcomes, Morbidity, and Mortality

Considering the low-risk status of the entire cohort, the peri- and postoperative courses proceeded mostly uneventfully in both treatment groups. MACCE, indicating the primary endpoints such as perioperative myocardial infarction (0.2% vs. 0.7%; *p* = 0.558), perioperative stroke (1.4% vs. 2.3%; *p* = 0.335), and 30-day mortality (1.4% vs. 0.9%; *p* = 0.558), were comparable between both groups (Figure 5). The incidence of perioperative disabling stroke (Rankin > 2) was even lower (0.6% vs. 0.9%; *p* = 0.824). The mortality rate in the MICLATS-AVR group was lower than the predicted mortality according to both EuroSCORE II (1.6 ± 1.1%) and STS-PROM Score (1.2 ± 0.7%). In contrast, the mortality rate in the RAT-AVR group closely matched its predicted mortality according to EuroSCORE II (1.6 ± 0.9%) and STS-PROM Score (1.4 ± 0.8%).

Additional postoperative major morbidities, such as respiratory failure, acute kidney injury requiring consecutive continuous veno-venous hemofiltration, and transient ischemic attack, were generally uncommon (Table 3). The distribution of their frequency did not differ significantly between the two treatment groups (Table 3). Postoperative delirium syndrome stood out as the most prevalent among the postoperative morbidities in both treatment groups. Nevertheless, no statistically significant differences were found between the groups concerning postoperative delirium (15.5% vs. 18.1%; *p* = 0.329). The ventilation time and the average number of packed red blood cells were comparable between both groups. While the length of ICU stay was comparable between the two treatment groups, patients in the MICLATS-AVR group experienced a slightly shorter yet statistically significant total length of hospital stay (9.3 ± 5.0 days vs. 9.7 ± 5.0 days; *p* = 0.003). The conversion rate to conventional sternotomy was numerically higher in the RAT-AVR group compared to the control group (3.7% vs. 1.6%; *p* = 0.069), although this difference only exhibited a statistical tendency and thus lacked significance. In contrast, re-exploration was more frequent in the MICLATS-AVR group (5.5% vs. 9.2%). The difference reached statistical significance (*p* = 0.04). A subgroup analysis of patients who underwent re-exploration revealed that 71.8% (*n* = 28) of re-explorations in the MICLATS-AVR group were performed due to postoperative bleeding, with just over a quarter of patients being re-explored due to postoperative increasing subcutaneous emphysema (*n* = 11; 28.2%). In most cases, the source of bleeding was access-related bleeding from the chest wall (*n/N* = 25/28; 89.3%) rather than bleeding from the cannulation sites or the aortotomy (*n/N* = 3/28; 10.7%). Postoperative bleeding was the main reason for re-exploration in the RAT-AVR group and accounted for most cases (*n/N* = 23/27; 85.2%). At 14.8% (*n/N* = 4/27), the incidence of subcutaneous emphysema was lower in the RAT-AVR group than in the MICLATS-AVR group. The source of bleeding in the RAT-AVR group was distributed almost evenly between access-related bleedings (*n/N* = 11/23; 47.8%) and bleedings from cannulation sites or aortotomy (*n/N* = 12/23; 52.2%). The incidence of postoperatively new-onset atrial fibrillation was similar in both groups. The postoperative implantation rate of a permanent pacemaker was numerically higher in the MICLATS-AVR group, although there was no statistical significance. Impaired wound healing was more prevalent in the RAT-AVR group than in the MICLATS-AVR group, reaching a significance level of *p* = 0.002. The subgroup analysis revealed that the leading site of impaired wound healing in the RAT-AVR group was the groin after cannulation of the femoral vessels (64.8%; *n/N* = 35/54). The remaining wound healing disorders occurred at the thoracic site of the surgical access route (35.2%; *n/N* = 19/54). Similarly, in the MICLATS-AVR group, impaired wound healing occurred more frequently at the groin (81.8%; *n/N* = 18/22) than at the site of the surgical access route (18.2%; *n/N* = 4/22). However, the proportion of thoracic wound healing abnormalities in the surgical access area was significantly higher in the RAT-AVR group (*p* = 0.034). In both groups, impaired wound healing was mainly due to the formation of lymphatic fistulas. A deep wound infection with consecutive mediastinitis occurred in one patient in the RAT-AVR group. Conduction disturbances requiring postoperative permanent pacemaker implantation occurred numerically more frequently in the MICLATS-AVR group than in the comparison group but showed no statistically significant difference. Postoperatively, new-onset atrial fibrillation exhibited a similar frequency distribution between both groups.

### 3.3. Propensity Score–Matched Cohort

To account for potential confounding variables, we conducted a propensity score-matched analysis between both groups, resulting in 718 patients (359 pairs) for subsequent analysis.

#### 3.3.1. Adjusted Procedural and Intraoperative Data

The findings for this section are outlined in Table 2. The Following propensity score matching, the MICLATS-AVR cohort solely revealed a notably extended duration of cardiopulmonary bypass (63.1 ± 20.4 vs. 66.4 ± 18.2; *p* ≤ 0.001) in comparison to the RAT-AVR group in terms of the surgical section times. Like the unadjusted outcomes, the implantation of rapid deployment valves in the MICLATS-AVR group was significantly more prevalent (81.3% vs. 53.6%; *p* ≤ 0.001; Figure 3)

#### 3.3.2. Adjusted Postoperative Outcomes, Morbidity, and Mortality

The results for this section are delineated in Table 2. In line with the unadjusted analysis, the MICLATS-AVR group demonstrated a marginally reduced but statistically significant overall hospitalization duration (9.2 ± 4.5 vs. 9.7 ± 5.2; *p* = 0.01). Further postoperative outcomes did not differ significantly between propensity-matched groups (Figure 6).

## 4. Discussion

For an extended period, the widely accepted standard approach to aortic valve surgery has been a complete median sternotomy despite the availability of various approaches to MICS-AVR. The field has witnessed a transformation, with surgical aortic valve replacement now amenable to minimally invasive techniques, such as sternum-sparing thoracotomies or partial hemisternotomies. Numerous studies comparing these techniques with conventional sternotomy have consistently demonstrated clear advantages, extending beyond anticipated results and cosmetically desirable outcomes [10,15,16,17]. This not only underscores the efficacy of minimally invasive approaches but also emphasizes the undeniable patient preference for methods that reduce discomfort, expedite recovery, and enhance cosmetic results.

Despite these advancements, a persistent debate surrounds how to effectively meet the growing demand for minimally invasive procedures. In Germany, the prevailing approach for aortic valve surgery continues to be full median sternotomy, while partial hemisternotomy stands out as the predominant minimally invasive method, accounting for 39.7% of cases. Intriguingly, sternum-sparing minimally invasive techniques, although promising, are conspicuously absent from the annual German Heart Surgery Report [6].

Moreover, driven by the interventional catheter-based therapeutic procedure of TAVI, the utilization of partial hemisternotomy as a minimally invasive treatment approach has steadily increased since 2007, when this proportion was limited to 4.7% [8,18]. However, the fact that this percentage has not surpassed the 50% mark over the years raises questions about the trajectory of modern cardiac surgery and the broader adoption of minimally invasive techniques. As part of the institutionalization and establishment of a program for minimally invasive surgical techniques for structural heart diseases, we implemented the innovative sternum-sparing approach of RAT-AVR, developed by Joseph Lamelas, in our institution in 2014 [19]. The streamlined right lateral transaxillary access, based on the surgical principle of ‘single incision—direct vision,’ as suggested by our team, was introduced in 2019 [9,11].

As mentioned above, numerous studies have offered comprehensive insights into the outcomes of MICS-AVR [20], most of them comparing sternum-sparing minimally invasive techniques with conventional full sternotomy or partial hemisternotomy. In this present study, to the best of our knowledge, we demonstrate, for the first time in a substantial patient cohort of 918 individuals, a comprehensive comparison of surgical outcomes between two sternum-sparing minimally invasive approaches. The key findings of this study can be outlined as follows:−No significant differences were observed between the two groups in terms of MACCE.−The MICLATS-AVR group exhibited a shorter overall hospital stay.−Significantly lower rates of postoperative impaired wound healing were noted in the MICLATS-AVR group.

The main adverse cardio-cerebral events, encompassing perioperative myocardial infarction, perioperative ischemic stroke, and 30-day mortality—defined as the primary endpoints of this study—revealed no significant differences between the two treatment groups. This suggests that the MICLATS-AVR procedure can be performed as a minimally invasive sternum-sparing alternative surgical approach just as safely as the already well-established RAT-AVR procedure.

There is considerable controversy in the literature regarding the heightened risk of perioperative stroke associated with femoral cannulation due to retrograde blood flow. The prevailing opinion suggests that femoral cannulation increases the risk of stroke during minimally invasive procedures [21,22,23,24,25]. The perioperative stroke rates in the two study groups were 1.4% and 2.2%, respectively, which is notably below the incidence of 4.5% to 5% reported in previous studies [22,24].

Hospital resource utilization, including length of ICU stay and ventilation time, was comparable between both groups, except for the overall hospital length of stay, which was significantly shorter in the MICLATS-AVR group. A possible explanation is that early patient mobilization and achieving patient independence due to accelerated recovery may be facilitated by the MICLATS technique, as it is bone-sparing and does not involve the shoulder girdle. Consequently, patients may be discharged earlier. However, this is a hypothesis that needs further investigation in subsequent studies.

There were no significant differences in major postoperative complications, including acute kidney injury, respiratory failure, or transient ischemic attack, in this series. However, postoperative delirium syndrome emerged as the most frequent postoperative complication in both treatment groups, although the difference between the groups was not statistically significant. This observation can be attributed to several factors. When discussed solely in the context of the cardiac surgical approach, the generalized assertion can be made that the notable incidence rate of postoperative delirium syndrome may be attributed, in part, to the restricted feasibility of the de-airing procedure or the retrograde flow in femoral cannulation in minimally invasive cardiac surgeries. However, this explanation is highly likely to be insufficient to account for this phenomenon. Previously, we examined the feasibility of MICLATS-AVR compared to full conventional sternotomy, in which we also observed a comparable frequency of postoperative delirium syndrome (15.9%) in the sternotomy group, which did not significantly differ from that in the MICLATS-AVR group [10]. Therefore, the cause of the development of postoperative delirium syndrome appears to be also influenced by other circumstances that were not the primary focus of this study, as they do not align with the primary research question.

The higher frequency of re-exploration, while not statistically significant after propensity matching, is a shortcoming that should be addressed. An analysis of the frequency of re-explorations over the years showed a decline. This decline suggests an initial learning curve.

Regarding postoperative wound healing abnormalities, the existing literature provides clear indications that RAT-AVR can result in a lower incidence rate compared to sternotomy and partial hemisternotomy [20,26,27,28,29]. The incidence of thoracic wound healing abnormalities was notably higher in the RAT-AVR group in comparison to the MICLATS-AVR group. One plausible explanation for this disparity may be rooted in the fact that, despite being sternum-sparing, the RAT-AVR technique is not bone-sparing since the detachment of the third rib from the sternum is required. Furthermore, the local blood supply might face compromise due to the transection and detachment of the right mammary artery. These two access-related drawbacks could contribute to the more frequent occurrence of local wound healing abnormalities in the surgical access area in RAT-AVR.

In addition to the comparative investigation of this study, it should be noted that the MICLATS-AVR approach has certain advantages. Firstly, this approach is not only sternum-sparing but also bone-sparing, as the intercostal exposure is sufficient for the feasibility of the surgical procedure, preserving the right mammary artery. Secondly, combination procedures, such as multiple valve surgeries, can be easily performed through the MICLATS-AVR approach [30].

## 5. Conclusions

In our comprehensive comparison of the outcomes between MICLATS-AVR and RAT-AVR in a substantial cohort, we found that MICLATS-AVR is at least as safe as RAT-AVR and can be performed in the same time frame. No significant differences in MACCE were detected between both treatment groups. Notably, the MICLATS-AVR group exhibited a shorter hospital stay and lower rates of postoperative impaired wound healing, suggesting its feasibility and safety as an alternative to the well-established RAT-AVR.

However, beyond the results, this study serves as an inaugural exploration into the realm of sternum-sparing minimally invasive surgical access routes for isolated aortic valve replacement. It propels us to contemplate a broader perspective, challenging the contemporary paradigm of choosing between sternotomy or partial hemisternotomy in the era of TAVI. As we navigate the complexities of modern cardiac surgery, this study prompts a critical examination of the choices we make in meeting patients’ desires for less invasive procedures.

In addressing the current challenges of our time, this study underscores the imperative to expand the evidence base for sternum-sparing minimally invasive techniques. Despite the scarcity of data and evidence surrounding these procedures, our study contributes valuable insights, affirming the feasibility of both approaches and their capacity to yield satisfactory results. Notably, the MICLATS-AVR procedure demonstrates certain potential advantages, warranting further exploration.

In essence, this study marks the commencement of a discourse that delves into the intricacies of sternum-sparing approaches, paving the way for a more informed and expansive understanding of their role in contemporary cardiac surgery.

## 6. Limitations

This study carries certain limitations. Primarily, despite its extensive cohort, it remains a single-center retrospective study with a limited short-term follow-up. Secondly, the propensity-score-matching model used may not have accounted for unknown yet potentially relevant risk factors and confounders. An additional concern stems from the fact that matching parameters were primarily chosen based on surgical feasibility in minimally invasive aortic valve replacement procedures. Furthermore, it is crucial to note that the results achieved in minimally invasive aortic valve replacement were within a high-volume expert center and may not be generalizable to all patient populations.

## Figures and Tables

**Figure 1 jcm-13-00985-f001:**
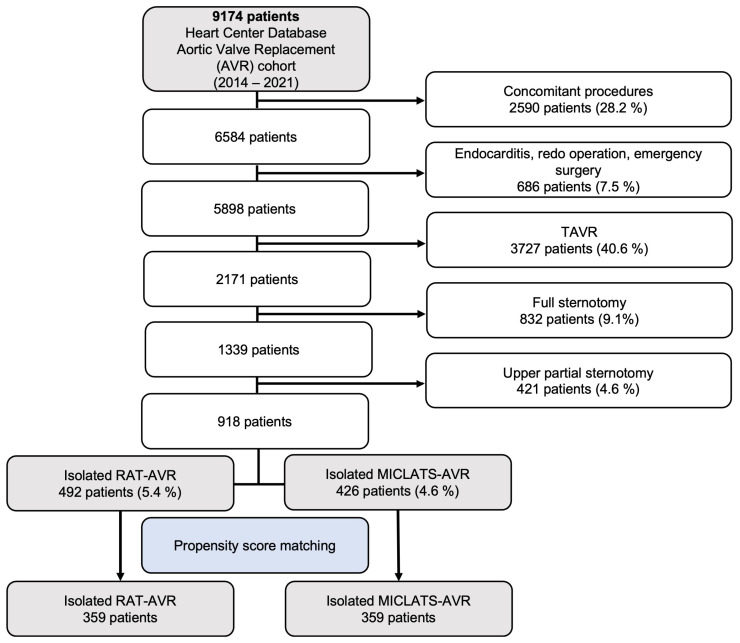
Flow diagram of the study population. Abbreviations: TAVR, trans-catheter aortic valve replacement; RAT-AVR, aortic valve replacement via right anterolateral thoracotomy; MICLATS-AVR, aortic valve replacement via right transaxillary thoracotomy.

**Figure 2 jcm-13-00985-f002:**
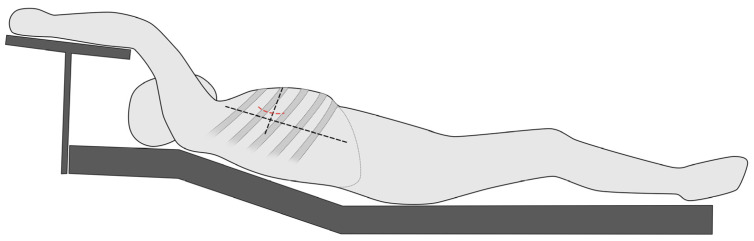
Schematic illustration regarding the positioning of the patient in the so-called spear-throwing position.

**Figure 3 jcm-13-00985-f003:**
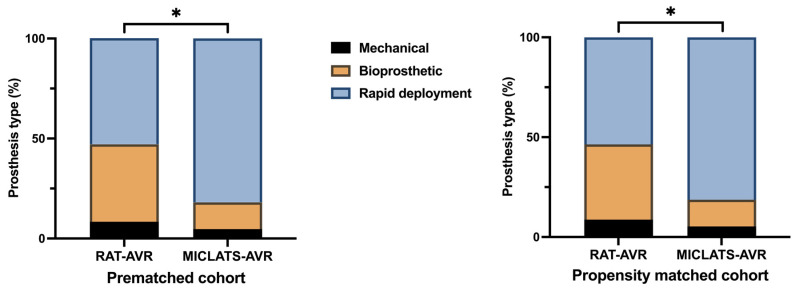
Distribution of aortic valve prosthesis type implanted within each treatment group. Note: * *p* < 0.05; because of the process of mathematical rounding, wherein percentages are rounded up or down to the nearest tenth decimal place, minute deviations of up to 0.1% from the absolute value of 100% can potentially manifest. Abbreviations: RAT-AVR, aortic valve replacement via right anterolateral thoracotomy; MICLATS-AVR, aortic valve replacement via right transaxillary thoracotomy.

**Figure 4 jcm-13-00985-f004:**
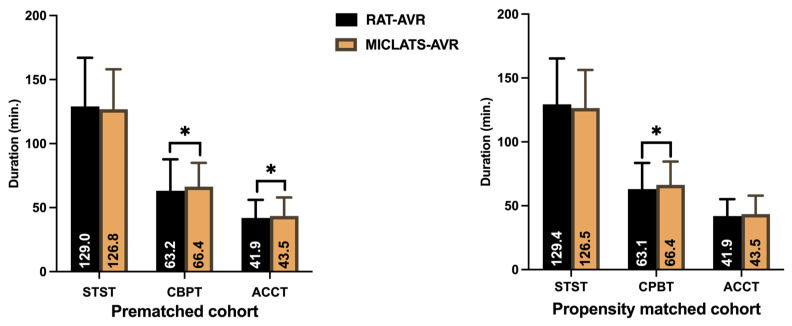
Surgical section times. Note: * *p* < 0.05 between groups. Abbreviations: RAT-AVR, aortic valve replacement via right anterolateral thoracotomy; MICLATS-AVR, aortic valve replacement via right transaxillary thoracotomy; min, minutes; STST, skin-to-skin time; CPBT, cardiopulmonary bypass time; ACCT, aortic cross-clamp time.

**Figure 5 jcm-13-00985-f005:**
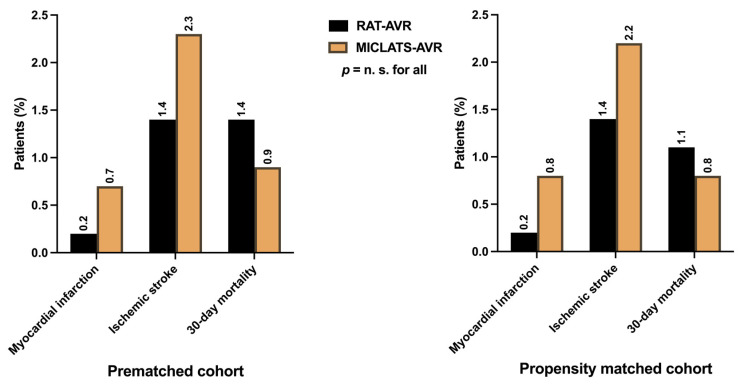
Major adverse cardio-cerebral events, defined as primary endpoints, including perioperative myocardial infarction, perioperative ischemic stroke, and 30-day mortality. Note: The depicted perioperative stroke event is completely independent of clinical symptom severity or graduation according to the modified Rankin scale. Abbreviations: RAT-AVR, aortic valve replacement via right anterolateral thoracotomy; MICLATS-AVR, aortic valve replacement via right transaxillary thoracotomy.

**Figure 6 jcm-13-00985-f006:**
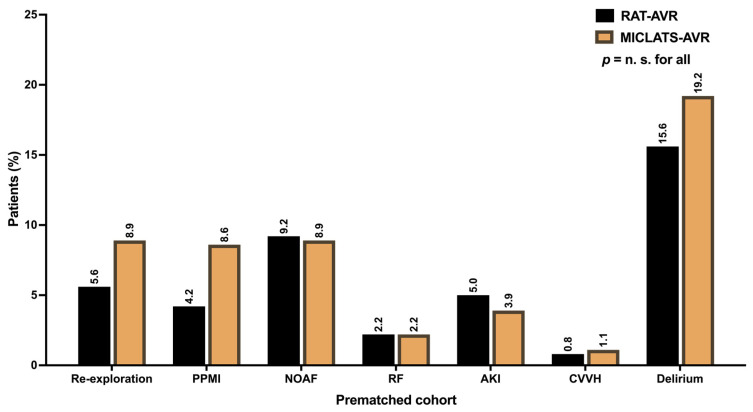
Postoperative outcomes. Abbreviation: RAT-AVR, aortic valve replacement via right anterolateral thoracotomy; MICLATS-AVR, aortic valve replacement via right transaxillary thoracotomy; PPMI, permanent pacemaker implantation; NOAF, new-onset atrial fibrillation; RF, respiratory failure; AKI, acute kidney injury; CVVH, continuous veno-venous hemofiltration.

**Table 1 jcm-13-00985-t001:** Baseline characteristics.

	Pre-Matched Cohort			Propensity Score-Matched Cohort		
	RAT-AVR (*n* = 492)	MICLATS-AVR (*n* = 426)	*p*	RAT-AVR (*n* = 359)	MICLATS-AVR (*n* = 359)	*p*
**Age (years), mean ± SD**	68.0 ± 10.0	68.0 ± 8.5	0.191	67.5 ± 9.7	67.6 ± 8.8	0.715
**Sex (male), n (%)**	312 (63.4)	264 (62.0)	0.681	234 (65.2)	228 (63.5)	0.697
**Height (cm), mean ± SD**	170.7 ± 9.4	170.7 ± 9.3	0.872	171.4 ± 9.5	170.7 ± 9.3	0.300
**Weight (kg), mean ± SD**	78.4 ± 13.6	81.7 ± 15.3	*≤0.001 ***	80.5 ± 13.7	80.6 ± 14.5	0.721
**BMI (kg/m^2^), mean ± SD**	26.8 ± 3.9	28.0 ± 4.6	*≤0.001 ***	27.4 ± 4.0	27.6 ± 4.3	0.456
**Arterial hypertension, *n* (%)**	453 (92.1)	380 (89.2)	0.139	333 (92.8)	320 (89.1)	0.118
**Diabetes mellitus, *n* (%)**	132 (26.8)	119 (27.9)	0.711	86 (24.0)	92 (25.6)	0.666
**Dyslipidemia, *n* (%)**	260 (52.8)	247 (58.0)	0.126	201 (56.0)	207 (57.7)	0.706
**Coronary artery disease, *n* (%)**	131 (26.6)	115 (27.0)	0.940	100 (27.9)	89 (24.8)	0.383
**LVEF (%), mean ± SD**	57.8 ± 10.5	56.9 ± 11.0	0.314	55.6 ± 8.5	56.0 ± 9.3	0.989
**COPD, *n* (%)**	38 (7.7)	30 (7.0)	0.707	20 (5.6)	26 (7.2)	0.446
**Pulmonary arterial hypertension, *n* (%)**	69 (14.0)	54 (12.7)	0.562	69 (14.0)	54 (12.7)	≥0.999
**Renal insufficiency, *n* (%)**	75 (15.2)	75 (17.6)	0.371	49 (13.6)	64 (17.8)	0.151
**Hemodialysis, *n* (%)**	2 (0.4)	3 (0.7)	0.668	0 (0.0)	3 (0.8)	0.249
**CRCL (mL/min.), mean ± SD**	80.8 ± 25.8	84.1 ± 25.9	*0.029 **	84.0 ± 25.6	83.5 ± 25.6	0.917
**PAOD, *n* (%)**	15 (3.0)	21 (4.9)	0.173	14 (3.9)	13 (3.6)	≥0.999
**Carotid artery stenosis > 50%, *n* (%)**	19 (3.9)	17 (4.0)	≥0.999	15 (4,2)	11 (3.1)	0.55
**TIA, *n* (%)**	14 (2.8)	5 (1.2)	≥0.999	12 (3.3)	5 (1.4)	0.139
**Ischemic stroke, *n* (%)**	20 (4.0)	21 (5.9)	0.543	13 (3.6)	15 (4.1)	0.714
**Pacemaker, *n* (%)**	13 (2.6)	17 (4.0)	0.269	6 (1.7)	10 (2.8)	0.449
**Smoker status, *n* (%)**	51 (10.4)	50 (11.7)	0.527	36 (10.0)	47 (13.1)	0.243
**NYHA class III or IV, *n* (%)**	259 (52.6)	280 (65.7)	*≤0.001 ***	220 (61.3)	223 (62.1)	0.803
**EuroSCORE II (%), mean ± SD**	1.6 ± 1.0	1.6 ± 1.1	0.764	1.6 ± 1.0	1.6 ± 1.1	0.632
STS-PROM Score, mean ± SD	1.4 ± 0.8	1.2 ± 0.7	*≤0.001 ***	1.2 ± 0.6	1.2 ± 0.7	0.125

Note: Bold and italic values indicate statistical significance: *, *p* ≤ 0.05; **, *p* ≤ 0.01. Abbreviations: RAT-AVR, aortic valve replacement via right anterolateral thoracotomy; MICLATS-AVR, aortic valve replacement via right transaxillary thoracotomy; SD, standard deviation; BMI, body mass index, LVEF, left ventricular ejection fraction; COPD, chronic obstructive pulmonary disease; CRCL, calculated creatinine-clearance according to the Cockcroft-Gault equation; PAOD, peripheral arterial occlusion disease; TIA, transient ischemic attack; NYHA, New York Heart Association; STS-PROM, Society of Thoracic Surgeons predicted risk of mortality.

**Table 2 jcm-13-00985-t002:** Procedural and intraoperative data.

	Pre-Matched Cohort			Propensity-Score-Matched Cohort		
	RAT-AVR (*n* = 492)	MICLATS-AVR (*n* = 426)	*p*	RAT-AVR (*n* = 359)	MICLATS-AVR (*n* = 359)	*p*
Prosthesis size (mm), mean ± SD	24.0 ± 1.9	24.2 ± 2.1	*0.044 ******	24.1 ± 1.9	24.2 ± 2.1	0.200
STST (min.), mean ± SD	129.0 ± 38.1	126.8 ± 31.2	0.845	129.4 ± 35.9	126.5 ± 29.8	0.790
CPBT (min.), mean ± SD	63.2 ± 24.5	66.4 ± 18.5	*≤0.001 *******	63.1 ± 20.4	66.4 ± 18.2	** *≤0.001 *** **
ACCT (min.), mean ± SD	41.9 ± 14.2	43.5 ± 14.4	*0.044 ******	41.9 ± 13.3	43.5 ± 14.4	0.182
Prosthesis type						
− Mechanical, *n* (%)	41 (8.4)	20 (4.7)		31 (8.7)	19 (5.3)	
− Bioprosthetic, *n* (%)	190 (38.7)	57 (13.4)	*≤0.001 ***	135 (37.7)	48 (13.4)	** *≤0.001 *** **
− RDV, *n* (%)	260 (53.0)	349 (81.9)		192 (53.6)	292 (81.3)	

Note: Bold and italic values indicate statistical significance: *, *p* ≤ 0.05; **, *p* ≤ 0.01. Abbreviations: RAT-AVR, aortic valve replacement via right anterolateral thoracotomy; MICLATS-AVR, aortic valve replacement via right transaxillary thoracotomy; SD, standard deviation; min, minutes; STST, skin-to-skin time; CPBT, cardiopulmonary bypass time; ACCT, aortic cross-clamp time; RDV, rapid deployment bioprosthetic valve.

**Table 3 jcm-13-00985-t003:** Postoperative morbidity and mortality.

	Pre-Matched Cohort			Propensity-Score-Matched Cohort		
	RAT-AVR (*n* = 492)	MICLATS-AVR (*n* = 426)	*p*	RAT-AVR (*n* = 359)	MICLATS-AVR (*n* = 359)	*p*
Ventilation time (hours) − ≤12, *n* (%) − ≤24, *n* (%) − >24, *n* (%)	447 (91.2) 31 (6.3) 12 (2.4)	382 (89.7) 32 (7.5) 12 (2.8)	0.426	326 (91.1) 21 (5.9) 11 (3.1)	321 (89.4) 29 (8.1) 9 (2.5)	0.487
Respiratory failure ^†^, *n* (%)	9 (1.8)	11 (2.6)	0.501	8 (2.2)	8 (2.2)	≥0.999
ICU stay (days), mean ± SD	1.9 ± 2.3	2.0 ± 3.2	0.833	1.9 ± 2.5	2.0 ± 2.7	0.465
Hospital stay (days), mean ± SD	9.7 ± 5.0	9.3 ± 5.0	*0.003 ***	9.7 ± 5.2	9.2 ± 4.5	** *0.01 ** **
Transfusion of PRBC, mean ± SD	0.6 ± 2.6	0.6 ± 1.6	0.358	0.6 ± 2.9	0.5 ± 1.4	0.067
AKI, *n* (%)	31 (6.3)	17 (4.0)	0.137	18 (5.0)	14 (3.9)	0.476
AKI grade II or III, *n* (%)	26 (5.3)	17 (4.0)	0.434	17 (4.7)	14 (3.9)	0.588
CVVH, *n* (%)	5 (1.0)	6 (1.4)	0.763	3 (0.8)	4 (1.1)	≥0.999
Conversion to sternotomy, *n* (%)	18 (3.7)	7 (1.6)	0.069	14 (3.9)	6 (1.7)	0.110
Re-exploration, *n* (%)	27 (5.5)	39 (9.2)	*0.040 **	20 (5.6)	32 (8.9)	0.112
Impaired wound healing, *n* (%)	54 (11.0)	22 (5.2)	*0.002 ***	42 (11.7)	14 (3.9)	** *<0.001 *** **
Postoperative delirium, *n* (%)	76 (15.5)	77 (18.1)	0.329	56 (15.6)	69 (19.2)	0.238
Ischemic stroke, *n* (%) any Rankin > 2	7 (1.4) 3 (0.6)	10 (2.3) 4 (0.9)	0.335 0.824	5 (1.4) 3 (0.8)	8 (2.2) 3 (0.8)	0.578 ≥0.999
TIA, *n* (%)	4 (0.8)	1 (0.2)	0.38	3 (0.8)	1 (0.3)	0.373
PPM implantation, *n* (%)	23 (4.7)	34 (8.0)	0.126	15 (4.2)	31 (8.6)	0.058
NOAF, *n* (%)	49 (10.0)	41 (9.6)	0.921	33 (9.2)	32 (8.9)	0.897
Myocardial infarction, *n* (%)	1 (0.2)	3 (0.7)	0.343	1 (0.3)	3 (0.8)	0.624
30-day mortality, *n* (%)	7 (1.4)	4 (0.9)	0.558	4 (1.1)	3 (0.8)	≥0.999

Note: Bold and italic values indicate statistical significance: *, *p* ≤ 0.05; ** *p* ≤ 0.01; ^†^, Defined as primary postoperative ventilation time ≥72 h, reintubation, and tracheotomy. Abbreviations: RAT-AVR, aortic valve replacement via right anterolateral thoracotomy; MICLATS-AVR, aortic valve replacement via right transaxillary thoracotomy; SD, standard deviation; ICU, intensive care unit; PRBC, packed red blood cells; AKI, acute kidney injury; CVVH, consecutive renal failure needing continuous veno-venous hemofiltration; TIA, transient ischemic attack; PPM, permanent pacemaker; NOAF, new-onset atrial fibrillation.

## Data Availability

The data presented in this study are available upon request from the corresponding author. The data are not publicly available due to ethical regulations.

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
