# Peer review of "Right Anterior versus Right Transaxillary Access for Minimally Invasive Aortic Valve Replacement: A Propensity Matched Competitive Analysis"

_jcm, 2024, doi:10.3390/jcm13040985_

Round 1
Reviewer 1 Report
Comments and Suggestions for Authors
The work comparing the various minimally invasive procedures for aortic valve replacement is highly relevant and makes an important contribution to the topic of surgical aortic valve replacement. The article is well written
There is no significance for MACCE. The significant results for length of stay and cardiopulmonary bypass time are less clinically relevant at 9.7 vs 9.3 days and 63.1 vs 66.4 minutes, respectively. The lower rate of wound healing disorders seems to be more decisive for the MICLATS group.
Overall, this is a very interesting study.
Author Response
Dear Reviewer 1,
Thank you for your thoughtful review and positive feedback on our work comparing various minimally invasive sternum sparing procedures for aortic valve replacement. We appreciate your acknowledgment of the relevance and contribution of our study to the field of surgical aortic valve replacement.
Your observation regarding the lower rate of wound healing disorders being more decisive for the MICLATS group aligns with our interpretation of the results.
We are pleased to hear that you find our study interesting.
Once again, we appreciate your time and thoughtful evaluation of our work.
With best regards,
Ali Taghizadeh-Waghefi
Reviewer 2 Report
Comments and Suggestions for Authors
The report is well written and interesting with good level of english. A comparison of surgical outcomes between two sternum-sparing minimally invasive approaches is performed. So, the MICLATS-AVR procedure can be performed as an alternative of already consolidated RAT-AVR procedure.
It could be improved by adding some details as the numbers of different surgeons performing the procedures and if the single experience had an impact on results.
Author Response
Dear Reviewer 2,
Thank you for your positive feedback on our report and for recognizing the quality of the writing and the level of English. We appreciate your insightful comments regarding the comparison of surgical outcomes between the two sternum-sparing minimally invasive approaches, MICLATS-AVR and RAT-AVR.
Your suggestion to include details such as the numbers of different surgeons performing the procedures and whether the single experience had an impact on results is well taken. We agree that providing this additional information could enhance the comprehensiveness of our study. Thus, we added the following paragraph on page 6, line 107 to 116:
"
2.4. Involved Surgeons
All MICS-AVR procedures were primarily conducted by four surgeons, each possessing expertise in the field of adult cardiac surgery, with all-over procedural volumes exceeding > 500, > 2,500, > 3,000, and > 6,500 cases involving ECC, respectively.
Additionally, a structured modular training approach was implemented for residents undergoing systematic step-by-step instruction in specific components of the MICS-AVR procedure. This encompasses training on femoral cannulation techniques, principles of thoracotomy, aspects of thoracotomy closure, and exposure of the aortic valve through various aortotomy techniques, adapted to the specific type of valve prosthesis to be implanted. Thus, in line with the institutionalization and conceptualization of the MICS-AVR techniques, residents are systematically assigned specific procedural components in a modular manner. This modular concept is designed to afford residents practical experience, enabling them to progressively undertake the operation themselves, initially under the supervision of a consultant and eventually autonomously."
We are grateful for your constructive input, and we believe that incorporating these details will contribute to the overall strength of the paper.
Thank you again for your valuable feedback.
With best regards,
Ali Taghizadeh-Waghefi
Reviewer 3 Report
Comments and Suggestions for Authors
This reviewer has the following comments to the Authors:
- Page 1, line 17: please correct RAT-AVR and not RAT-VAR
- References number 2 and 3 could be updated to the latest TAVI studies in patients with low risk (PARTNER 3 and Evolut Low Risk). [1-2]
- Page 3, line 109: It should be electrocardiography-gated computed tomography angiography.
- Page 4, lines 136-137: pay attention, there is two times the same sentence.
- Page 4, lines 143-145: please write better this sentence, it is not very clear. Moreover, add some references for the mentioned growing body of evidences.
- Page 10, lines 319-324: please add some references about all the evidence cited in these lines.
- Page 11, lines 325-331: please put reference number 6 at the end of this paragraph.
- Page 12, lines 398-403: in all this paragraph it should be RAT-AVR instead of MICLATS-AVR, because it is in the first that there is detachment of third rib and right internal mammary artery. There is confusion, please pay attention.
- All the numbers of citation should be written after the dot of the cited sentence.
References:
1. Forrest JK, Deeb GM, Yakubov SJ, et al. 3-Year Outcomes After Transcatheter or Surgical Aortic Valve Replacement in Low-Risk Patients With Aortic Stenosis. J Am Coll Cardiol. 2023;81(17):1663-1674. doi:10.1016/j.jacc.2023.02.017
2. Mack MJ, Leon MB, Thourani VH, et al. Transcatheter Aortic-Valve Replacement in Low-Risk Patients at Five Years. N Engl J Med. 2023;389(21):1949-1960. doi:10.1056/NEJMoa2307447
Author Response
Dear Reviewer 3,
Thank you for your detailed and insightful comments on our manuscript. We appreciate your thorough review and constructive suggestions. We would like to inform you that we have already addressed the points you raised:
-
Page 1, line 17: We have corrected "RAT-VAR" to "RAT-AVR" as per your suggestion.
-
References number 2 and 3: We have updated references 2 and 3 to include the latest TAVI studies in patients with low risk, specifically PARTNER 3 and Evolut Low Risk.
-
Page 3, line 109: We have corrected "electrocardiography-gated computed tomography angiography" as suggested.
-
Page 4, lines 136-137: We have rectified the repetition of the same sentence and made the necessary modifications.
-
Page 4, lines 143-145: The sentence has been rewritten for clarity, and references for the mentioned growing body of evidence have been added.
-
Page 10, lines 319-324: References have been added to support the evidence cited in these lines.
-
Page 11, lines 325-331: Reference number 6 has been moved to the end of this paragraph, as suggested.
-
Page 12, lines 398-403: The confusion between RAT-AVR and MICLATS-AVR has been corrected, ensuring consistency throughout the paragraph.
We appreciate your diligence in reviewing our manuscript, and we believe that these revisions significantly contribute to improving the quality of our work. If you have any further recommendations or concerns, please feel free to let us know.
Thank you again for your valuable feedback.
Sincerely,
Ali Taghizadeh-Waghefi